# Inhibition of Cyclin-Dependent Kinases 8/19 Restricts Bacterial and Virus-Induced Inflammatory Responses in Monocytes

**DOI:** 10.3390/v15061292

**Published:** 2023-05-31

**Authors:** Elena K. Kokinos, Sergey A. Tsymbal, Anastasia V. Galochkina, Svetlana A. Bezlepkina, Julia V. Nikolaeva, Sofia O. Vershinina, Anna A. Shtro, Victor V. Tatarskiy, Alexander A. Shtil, Eugenia V. Broude, Igor B. Roninson, Marina Dukhinova

**Affiliations:** 1SCAMT Institute, ITMO University, 9 Lomonosova Street, 191002 Saint-Petersburg, Russia; kokinos@scamt-itmo.ru (E.K.K.);; 2Smorodintsev Research Institute of Influenza, 15/17 Prof. Popov Street, 197376 Saint-Petersburg, Russia; 3Institute of Gene Biology, Russian Academy of Sciences, 34/5 Vavilova Street, 119334 Moscow, Russia; 4Blokhin National Medical Research Center of Oncology, Kashirskoe Highway 24, 115478 Moscow, Russia; 5Department of Drug Discovery and Biomedical Sciences, University of South Carolina, Sumter Street 715, Columbia, SC 29208, USA

**Keywords:** inflammation, monocytes, influenza virus, lipopolysaccharide, CDK8/19, cytokines

## Abstract

Hyperactivation of the immune system remains a dramatic, life-threatening complication of viral and bacterial infections, particularly during pneumonia. Therapeutic approaches to counteract local and systemic outbreaks of cytokine storm and to prevent tissue damage remain limited. Cyclin-dependent kinases 8 and 19 (CDK8/19) potentiate transcriptional responses to the altered microenvironment, but CDK8/19 potential in immunoregulation is not fully understood. In the present study, we investigated how a selective CDK8/19 inhibitor, Senexin B, impacts the immunogenic profiles of monocytic cells stimulated using influenza virus H1N1 or bacterial lipopolysaccharides. Senexin B was able to prevent the induction of gene expression of proinflammatory cytokines in THP1 and U937 cell lines and in human peripheral blood-derived mononuclear cells. Moreover, Senexin B substantially reduced functional manifestations of inflammation, including clustering and chemokine-dependent migration of THP1 monocytes and human pulmonary fibroblasts (HPF).

## 1. Introduction

Cytokine storm, a characteristic feature of immune over-reactivity that occurs in viral and bacterial pneumonia, is associated with high rates of disability and mortality worldwide [1]. Proinflammatory cytokines, such as interleukin (IL)-6, interleukin (IL)-1β, TNF-a, IFN-γ, and various chemokines, are involved in systemic inflammation, tissue damage and multiple organ failure. These functional peptides are produced and released by several cell types including monocytes and macrophages, natural killer cells, dendritic cells, T-cells, epithelial cells, and fibroblasts [2]. Transcriptional activation of cytokine expression is driven via Toll-like or cytokine receptor signaling and is controlled by the transcription factors NFκB, STAT1, STAT3, and others [3]. Progressive cytokine release, which is assisted by monocyte transendothelial migration, facilitates further activation of innate and adaptive immunity.

Recently, it was shown that transcription driven by external stimulation relies on transcriptional factors together with other coactivators, such as cyclin-dependent kinases (CDK) 8 and 19. Unlike most of the members of CDK family, CDK8 and its paralog, CDK19, do not participate in cell cycle regulation but play an important role in fine adjustment of cell transcriptional profile and cell adaptation to microenvironment [4]. In an interchangeable manner, CDK8 and CDK19 together with cyclin C (CycC), MED12, and MED13 comprise the mediator kinase module. This module regulates transcription via reversible association with the mediator complex, a multiprotein structure transducing regulatory signals from transcription factors to RNA polymerase II [5]. Along with other transcriptional CDKs, CDK8/19 can phosphorylate (directly or indirectly) the C-terminal domain of RNA polymerase II enabling initiation, elongation, and RNA processing. Upon engagement of the novel incoming signal, CDK8/19 mediate the phosphorylation of RNA polymerase II, supporting prolongation and stabilization of its activity on mRNA transcription. However, unlike other transcriptional CDKs, CDK8/19 also can regulate transcription at a more precise level by phosphorylation of some sequence-specific transcription factors [6]. Therefore, CDK8/19 regulation of transcription is not general but rather specific to certain stimuli since it affects genes only in combination with specific transcription factors. In adult mammals, CDK8/19 play a minor role under normal physiological conditions, but their expression and functionality become upregulated in pathologies. Particularly, in cancer, CDK8/19 act as oncogenes, and their overexpression has been associated with several types of cancer, such as breast, colorectal, and colon cancer [7].

Of special relevance to inflammation, CDK8/19 potentiate the effects of several signal-inducible transcription factors including NFκB [8] and modulate the activity of STAT1 and STAT3 [6]. Recently, a number of studies showed that CDK8/19 could be associated with the polarization of immune cells towards proinflammatory profiles, through effects involving STAT1 [9], STAT3 [10], or NFκB [11]. Furthermore, in vitro and in vivo research suggest that CDK8/19 play a role in functionalization of immune cells, including NK cells and T lymphocytes in context of antitumor immunity and autoimmune disorders [12,13,14,15]. At the same time, the role of CDK8/19 in antiviral and antibacterial responses of monocytes and macrophages remains poorly investigated. In the present study, we address the roles of CDK8/19 in influenza virus (strain H1N1) and bacterial lipopolysaccharide (LPS)-driven inflammatory responses in THP1 and U937 monocytic cell lines and peripheral blood-derived mononuclear cells (PBMCs) and evaluate the potential of a selective CDK8/19 inhibitor, Senexin B [16,17], to control proinflammatory activity in vitro.

## 2. Materials and Methods

### 2.1. Cell Culture Studies

THP1 and U937 cell lines were obtained from American Type Culture Collection, Manassas, VA, USA. PBMCs were isolated from the peripheral blood of 4 healthy male donors using a Ficoll gradient as described [18,19]. Cells were cultured in RPMI-1640 (Biolot, Saint-Petersburg, Russia) supplemented using 20% fetal bovine serum (FBS) (GE Healthcare Life Sciences, Russia), 100 units/mL penicillin, and 100 µg/mL streptomycin (Biolot, Russia) at 37 °C, 5% CO_2_ in humidified atmosphere. Influenza A virus (A/Puerto Rico/8/34 (H1N1)) was obtained from the Collection of Influenza and SARS Viruses at Smorodintsev Research Institute of Influenza, Saint Petersburg, Russia. Virus propagation and all experimentation with the viral material were performed at the Laboratory of Chemotherapy of Viral Infections, Smorodintsev Research Institute of Influenza, Saint-Petersburg, Russia. Viral particles with total multiplicity of infection (MOI) of 0.1 or *E. coli* LPS (1 μg/mL) (Sigma, St. Louis, MI, USA) were added to cell cultures 24 h prior to the analysis.

### 2.2. Resazurin Cytotoxicity Assay

Cell proliferation effects of CDK8/19 inhibitor, Senexin B (Senex Biotechnology, Columbia, SC, USA), was evaluated using a resazurin assay. Briefly, THP1 cells or PBMCs (5000 cells per well) were seeded on a 96-well plate (Jet BioFil, Guangzhou, China) and incubated with the drug for 72 h. Resazurin solution (working concentration 100 µg/mL) was added 16 h before the measurements. The amount of the reduced resazurin form was quantified using a spectrofluorometer (Tecan Spark multimode microplate reader (Switzerland); 560/590 nm excitation/emission wavelength). Cell viability in control conditions was taken as 100%. Three biological experiments with 3 technical replicates were performed throughout the study; mean and standard deviation values are shown on the graphs.

### 2.3. Transwell Migration Assay

THP1 cells (5 × 10^5^ cells per mL) were incubated with 1 µM Senexin B for 1 h, then activated with LPS (2 μg/mL) and left for 12 h or 24 h before supernatant collection. To estimate the migration capacity of THP1 cells the collected supernatant was added into a 96-well Transwell plate (Corning, Somerville, MA, USA) containing porous inserts of 5 μm; the inserts were then allowed to moisten for 30 min. 5 × 10^4^ cells per well were incubated for 2 h; afterwards the cells that migrated to the bottom chamber were counted with CytoFLEX flow cytometer (Beckman Coulter, Brea, CA, USA).

### 2.4. Scratch Assay

Primary human pulmonary fibroblasts (HPF) (gift from E. Dashinimaev, Engelhardt Institute of Molecular Biology, Moscow, Russia) were seeded onto 6-well plates at 7.5 × 10^5^ cells per well and allowed to recover at 37 °C 5% CO_2_ for 24 h. Next day, cells were treated with Senexin B at 1 µM concentration, *E. coli* LPS (1 μg/mL) (Sigma, St. Louis, MI, USA), or Senexin B 1 h prior to LPS. Cells from control wells were left untreated. After 24 h of incubation monolayer of HPF was scratched using a 200 μL pipette tip. Micrographs of the wound were taken at several time points: after the scratch, 4, 8, 18, and 24 h after the scratch. At each time point, 3–5 micrographs of each well were taken. Three independent experiments with 3 repeats were performed. Analysis of the wound area was conducted using ImageJ 1.52u program with the Wound Healing Tool plugin. The average wound area of control was normalized to 1 for each experiment; then, the data were combined, and average values were counted for each group.

### 2.5. RNA Extraction, Reverse Transcription, and Quantitative Real-Time PCR

5 × 10^5^ cells (THP1, U937 or PBMCs) were seeded onto 6-well plates and allowed to recover for 24 h. CDK8/19 inhibitor Senexin B was added at 1 µM concentration 1 h prior to the exposure to virus or LPS. After 24 h incubation, total RNA was extracted from the cells with the ExtractRNA kit 31 (Evrogen, Moscow, Russia) according to the manufacturer’s recommendations. Quantitative and qualitative analysis of the extracted RNA was performed on the NanoDrop spectrophotometer (Thermo Fisher Scientific, Waltham, MA, USA). Reverse transcription mix was prepared with 2 µg of RNA per reaction and MMLV-RT kit (Evrogen, Moscow, Russia) according to the manufacturer’s protocol. Real-time quantitative PCR (qPCR) assays were performed with the CFX Connect™ Real-Time PCR Detection System (BioRad Laboratories, Hercules, CA, USA) using the qPCRmix-HS SYBR reagent and primer mixes (Evrogen, Moscow, Russia) and primer. The primer sequences are shown in Table 1. The HPRT1 gene was used for the normalization of RNA levels, and relative gene expression was calculated.

### 2.6. Western Blot Analysis

Western blotting was used for specific detection of NFκB, STAT1, and STAT3 and their phosphorylated forms. Cells were seeded in a 6-well plate (500,000 cells per well) and incubated for 24 h. After incubation cells were lysed in RIPA buffer (150 mM NaCl, 1% NP-40, 0.1% SDS, 50 mM Tris base pH 8.0, 2 mM phenylmethylsulfonyl fluoride) and incubated using protein inhibitor cocktail (Roche) for 30 min on ice. Bradford reagent was used to evaluate total protein concentration. Cell lysates containing 30 µg of protein were resolved using sulfate polyacrylamide gel (SDS-PAGE) electrophoresis (120 V, 90 min) and transferred onto 0.2 µm nitrocellulose membranes (GE Healthcare, Chicago, IL, USA) for staining with rabbit primary antibodies against total and phosphorylated NFκB (p65-Ser536), STAT1 (Ser727, Y701), STAT3 (Ser727) (Cell Signaling, Danvers, MA, USA), and with secondary antibodies conjugated with horseradish peroxidase (Cell Signaling, Danvers, MA, USA). Proteins were visualized using the enhanced chemiluminescence reagent and a ChemiDoc MP gel imaging system (BioRad, Hercules, CA, USA). Semiquantitative signal image analysis was performed using ImageJ (v. 1.53s) software.

### 2.7. Statistical Analysis

Statistical analysis was performed using GraphPad version 8.0.1 (GraphPad Software, San Diego, CA, USA). At least 3 independent experiments were performed for each assay. The data were analyzed using an unpaired two-tailed *t*-test or Mann–Whitney U-test, as specified. *p* values less than 0.05 were considered significant (* *p* < 0.05, ** *p* < 0.01, *** *p* < 0.001, **** *p* < 0.0001).

## 3. Results

### 3.1. CDK8/19 Inhibition Reduces the Induction of mRNA Levels of Key Proinflammatory Cytokines by Influenza Virus and LPS in Monocytic Cells

To evaluate the roles of CDK8/19 in the regulation of cytokine genes in monocytic cells, we used a selective CDK8/19 inhibitor, Senexin B [16,17]. Senexin B (1 µM) was added to monocyte-derived cell lines (THP1, U937) and PBMCs 1 h before exposure to influenza A virus (A/Puerto Rico/8/34 (H1N1)) or bacterial LPS (Figure 1). Of note, the chosen virus strain represents a common trigger of seasonal pneumonia in Russia and worldwide [20].

QRT-PCR analysis demonstrated that both the virus and LPS induced a significant increase in mRNA of proinflammatory cytokines IL6, IL1B, CCL2, CXCL10, and TNFα after 24 h treatment (Figure 1). PBMCs and THP1 are derived from the peripheral blood, U937 cells have a different tissue origin (pleural cavity) [21]. Accordingly, THP1 and PBMCs had higher expression levels of proinflammatory cytokines, particularly following LPS activation. In contrast, IL6 mRNA was not expressed in unprimed U937 cells, nor was it elevated by virus or LPS. Levels of mRNA of other tested cytokines were also lower in LPS- and virus-primed U937 cells compared to THP1 or PBMCs (Figure 1B).

We next studied the effect of CDK8/19 inhibition on the transcriptional levels of pro-inflammatory cytokines. Senexin B had little-to-no impact on cytokine gene expression in quiescent cells, as mRNA levels of IL6, CXCL10, and TNFα did not change (Figure 1). At the same time, IL1B and CCL2 were slightly reduced in THP1 and PBMCs and significantly reduced in U937 (Figure 1). Of interest, U937 are monocyte-like cells of tissue origin, while THP1 and PBMCs are less differentiated, blood-derived monocytes (Figure 1B). When cells were incubated with LPS or influenza virus to model proinflammatory conditions, pretreatment with Senexin B substantially reduced LPS- and virus-induced gene expression in all studied cell models. The response of activated THP1, U937, and PBMCs to the drug was similar between the studied cell models (Figure 1). As expected, Senexin B could efficiently prevent CDK8/19-dependent transcription in a broad range of cell types, not limited to those of myeloid lineage.

The mRNA levels of TNFα in THP1 and PBMCs were also evaluated. Gene expression levels of TNFα in PBMCs and THP1 were on the borderline of the detection limits. Thus, TNFα may not be suitable to characterize inflammatory responses in THP1 and PBMCs in the studied in vitro models and was not included in the further studies.

### 3.2. CDK8/19 Inhibitor Downregulates S727 Phosphorylation of STAT1 and Upregulates NFκB p65 S536 Phosphorylation

Based on the obtained results, we continued to investigate the effects of CDK8/19 inhibition in the THP1 cell line, which was considered a suitable model of inflammation with the cytokine gene expression profile similar to that of PBMCs (Figure 1A,C). We then asked whether Senexin B has any impact on phosphorylation of transcription factors, which have been linked to CDK8/19-dependent transcription. For that purpose, we performed Western blot analysis of THP1 cells activated with LPS or influenza A virus A/Puerto Rico/8/34 (H1N1) and pretreated with Senexin B. LPS increased the phosphorylation levels of STAT1 (Ser727) and NFκB (phospho-NFκB p65 (Ser536)) (Figure 2A,B, Appendix A). Senexin B itself did not have any impact on STAT1 or STAT3 phosphorylation levels. As expected, STAT1 Ser727 phosphorylation was substantially reduced in the presence of CDK8/19 inhibitor in proinflammatory conditions (Figure 2A,B). Interestingly, CDK8/19 inhibition increased the levels of phospho-p65 both in untreated and LPS- or virus-activated THP1 cells (Figure 2B) suggesting the differential roles of CDK8/19 in transcriptional regulation.

### 3.3. Reduced THP1 Cell Clustering and Migratory Activity in the Presence of CDK8/19 Inhibitor

As described above, CDK8/19 inhibition shaped the transcriptional machinery of activated monocytes, reducing the levels of proinflammatory cytokines induced both using influenza virus and LPS. Next, we addressed the role of CDK8/19 in monocyte functional manifestation during viral infection or LPS exposure.

Similar to many other viruses, influenza A virus (A/Puerto Rico/8/34 (H1N1)) caused cell clustering of THP1 cells (Figure 3B). Of note, formation of cell clusters is a mechanism that facilitates viral spread from the infected to the healthy cell and, therefore, is involved in disease pathogenesis [22,23]. Our visual observations showed that cell clustering, a common hallmark of viral infection, was almost completely suppressed by Senexin B treatment of THP1 cells (Figure 3A–C). We then compared the relative numbers of cell clusters in control conditions or after influenza virus and Senexin B treatment. The quantitative analysis revealed an almost 14-fold increase in cluster number in virus-challenged THP1 cells (Figure 3D). Cluster formation was completely abolished by pretreatment using CDK8/19 inhibitor Senexin B (Figure 3D).

Since cell migration is driven, among other factors, by chemokines including CXCL10 and CCL2, we also checked whether mobility of non-monocytic cells can be impacted by CDK8/19 inhibition. For that, we activated human fibroblasts with LPS and performed experimental scratch wound. Senexin B was used as a selective CDK8/19 inhibitor, and wound size was analyzed 12 and 24 h after the scratch. We found that, at the studied time points, the average wound area in control cells was similar to that in cells treated using Senexin B. On the other hand, cells treated using LPS had a more rapid rate of wound healing resulting in a smaller wound area (Figure 4A–E). The average wound area of cells treated with Senexin B 1 h prior to LPS treatment was higher than that of the cells from LPS-treated group, indicating that cells were not actively migrating to the site of the wound and the area of the scratch was not decreasing as rapidly as in LPS-treated cells (Figure 4E). Our data indicate that LPS activation of human pulmonary fibroblasts leads to increased migration activity to the site of wound. However, treatment with Senexin B led to a decrease in the migration activity of LPS-activated cells to values comparable to control cells (Figure 4E). These data are consistent with our assumption that reduced chemokine expression resulting from CDK8/19 inhibition affects migration activity of nearby cells of nonimmune nature, such as fibroblasts. Therefore, we can assume that CDK8/19 play an important role in the transcriptional regulation of chemokine expression not only in monocytic cells but also in fibroblasts.

Another route for pathogen delivery to organs and tissues is provided by transendothelial migration of activated monocytes [24]. In addition, migrating monocytes contribute to the progression of cytokine storm from local to systemic level and, ultimately, septic state [25,26], while cytokines themselves are important drivers of monocyte migration. Consequently, reduced cytokine production upon CDK8/19 inhibition may also restrict the migratory potential of the cells. To confirm this hypothesis, we performed a Transwell membrane migration test and compared cell migration driven by the conditioned medium collected from cells treated with LPS with or without Senexin B. As expected, Senexin B not only reduced mRNA levels of proinflammatory cytokines (see above), but also decreased the cell-derived soluble factors with chemotactic activities as the numbers of migrated cells were substantially reduced at 12 h and 24 h (Figure 4F).

Thus, our results indicate that CDK8/19 inhibition can significantly reduce the pro-inflammatory response manifesting in increased expression levels of proinflammatory cytokines and phosphorylation of inflammation-associated transcription factor STAT1 at Ser727. Furthermore, treatment with Senexin B decreased signs of the functional manifestation of inflammation by negatively influencing cluster formation in THP-1 monocytic cells and migration activity of human pulmonary fibroblasts.

### 3.4. CDK8/19 Inhibition Is Not Associated with Cytotoxic Responses in THP1 Monocytes and Peripheral Blood-Derived Mononuclear Cells

Finally, we evaluated the potential cytotoxic effects of CDK8/19 inhibitor Senexin B. Previous reports showed that CDK8/19 inhibition does not trigger cell death or severe side effects in vitro or in vivo, and CDK8/19 activities are not generally required for cell proliferation [27,28]. Accordingly, we found that cell viability of THP1 (Figure 5A) or PBMCs (Figure 5B) remained stable upon 72 h of Senexin B treatment. Even at the highest concentrations, CDK8/19 did not reduce cell viability in both models. Thus, Senexin B can potentially limit manifestations of inflammation associated with cytokine production with no substantial impact on the viability of monocytes.

It is worth noting that CDK8/19 inhibitors can increase cytotoxic activities of other drugs, and CDK8/19 is an important co-target in cancer therapy [29,30,31,32,33,34]. Our results provide preliminary evidence that CDK8/19 inhibition may regulate inflammatory profiles of tumor-associated macrophages. Therefore, the roles of CDK8/19 inhibitors in cancer chemotherapy and immunotherapy may require particular attention to the antitumor activities of macrophages.

## 4. Discussion

Targeting CDK8/19 can provide a potential strategy for selective downregulation of gene transcription induced by pathogenic alterations in the cell microenvironment and early-stage control of cytokine production and release. CDK8/19-based regulation was shown to be a therapeutic target in cancer cells [17,31,33,34] and to affect certain immunogenic activities [12,13,14,35].

In the present study, we focused on the roles of CDK8/19 in transcriptional control of cytokine production. We showed, for the first time, that CDK8/19 inhibition can substantially reduce the proinflammatory activity of monocytic cells stimulated using influenza virus or LPS. We found that virus-induced mRNA levels of key proinflammatory cytokines, IL6, IL1B, CCL2, and CXCL10, were decreased in monocytic cell lines, namely THP1 and U937, and in PBMCs treated with Senexin B prior to viral infection (Figure 1). IL1b, IL6, TNFα, CCL2, and CXCL10 were selected as key markers of the inflammatory response including regulation of hematopoiesis, immune cell recruitment, activation, and functionalization [36,37,38,39,40,41,42,43].

Of note, influenza A virus (A/Puerto Rico/8/34 (H1N1), which is used in this study, does not replicate in human monocytic cells but the interaction of such cells with the virus drives Toll-like receptor-mediated activation. Thus, the observed reduction in proinflammatory activities of THP1 or mononuclear cells is not due to the possible impact on viral replication and different viral burden levels but to the limited proinflammatory capacity of the host cells. Moreover, the type of initial stimuli does not impact the efficiency of cytokine downregulation in the cells by CDK8/19 inhibition, suggesting CDK8/19 as a common molecular mechanism of proinflammatory response in myeloid cells.

The cytokine production is controlled by a number of transcription factors, such as STAT1 and NFκB, and their activities can be further promoted using a positive feedback loop. We found that STAT1 and STAT3 Ser727 phosphorylation levels were lower in THP1 cells upon CDK8/19 inhibition, in agreement with previous reports (Figure 2) [6,9,10]. In contrast, CDK8/19 inhibition did not inhibit NFκB phosphorylation, and Senexin B even increased LPS-induced S536 phosphorylation of p65 subunit (Figure 2). This result is in agreement with the previous report that CDK8/19 inhibition does not diminish NFκB translocation into the nucleus but suppresses NFκB-induced transcription downstream of NFκB [8,11]. Altered transcription factor phosphorylation levels can be a direct effect of CDK8/19 inhibition, since CDK8/19 can phosphorylate a broad variety of targets, including STAT1 and STAT3 [6,10]. It is also possible that reduced activation of transcription factors is due to the lower cytokine production and self-activation of monocytic cells.

We were further interested in characterizing the functional profiles of monocytic cells exposed to the virus or LPS in the presence or absence of the CDK8/19 inhibitor. We observed that the CDK8/19 inhibitor prevented formation of THP1 cell clusters and may therefore restrict virus transmission. We also discovered that CDK8/19 inhibitor negatively influences migration activity of human pulmonary fibroblasts activated by LPS. We therefore suggest that CDK8/19 may play an essential role in the regulation of proinflammatory cytokine expression in other cells involved in inflammation, such as pulmonary fibroblasts. Together with the reports that CDK8/19 inhibition suppresses the replication of other viruses [44,45,46], our findings suggest that CDK8/19 inhibitors may be broadly beneficial for the treatment of different viral and bacterial infections and potential immunomodulators.

The current study aimed to explore CDK8/19-dependent immune responses in viral and bacterial infections. We used a selective CDK8/19 inhibitor, Senexin B, to evaluate the roles of CDK8/19 in monocyte-mediated acute inflammation, cytokine production, and cell functional responses. Importantly, CDK8/19 are broad-specificity drivers of viral and bacterial inflammation, and CDK8/19 inhibitor Senexin B can substantially restrict the immunoreactivity of monocytes in the proposed models while not affecting cell survival and baseline activities. The cellular models used in this study elucidate the roles of CDK8/19 in monocyte-mediated inflammation. Since we were specifically focused on the role of monocytes in infection, the present study does not address the other components of innate and adaptive immunity. At the same time, we found that CDK8/19 may also potentiate fibroblast migration in vitro, suggesting that long-term tissue reorganization may also involve CDK8/19. Further in vivo experiments are required to improve our understanding of CDK8/19 roles in other immune cell subsets and the general course of viral and bacterial inflammation. Previous studies showed lack of in vivo toxicity of CDK8/19 inhibitors in adult animals and suggest that Senexin B or new generation CDK8/19 inhibitors can be used to evaluate the potential new approach against cytokine storm.

## Figures and Tables

**Figure 1 viruses-15-01292-f001:**
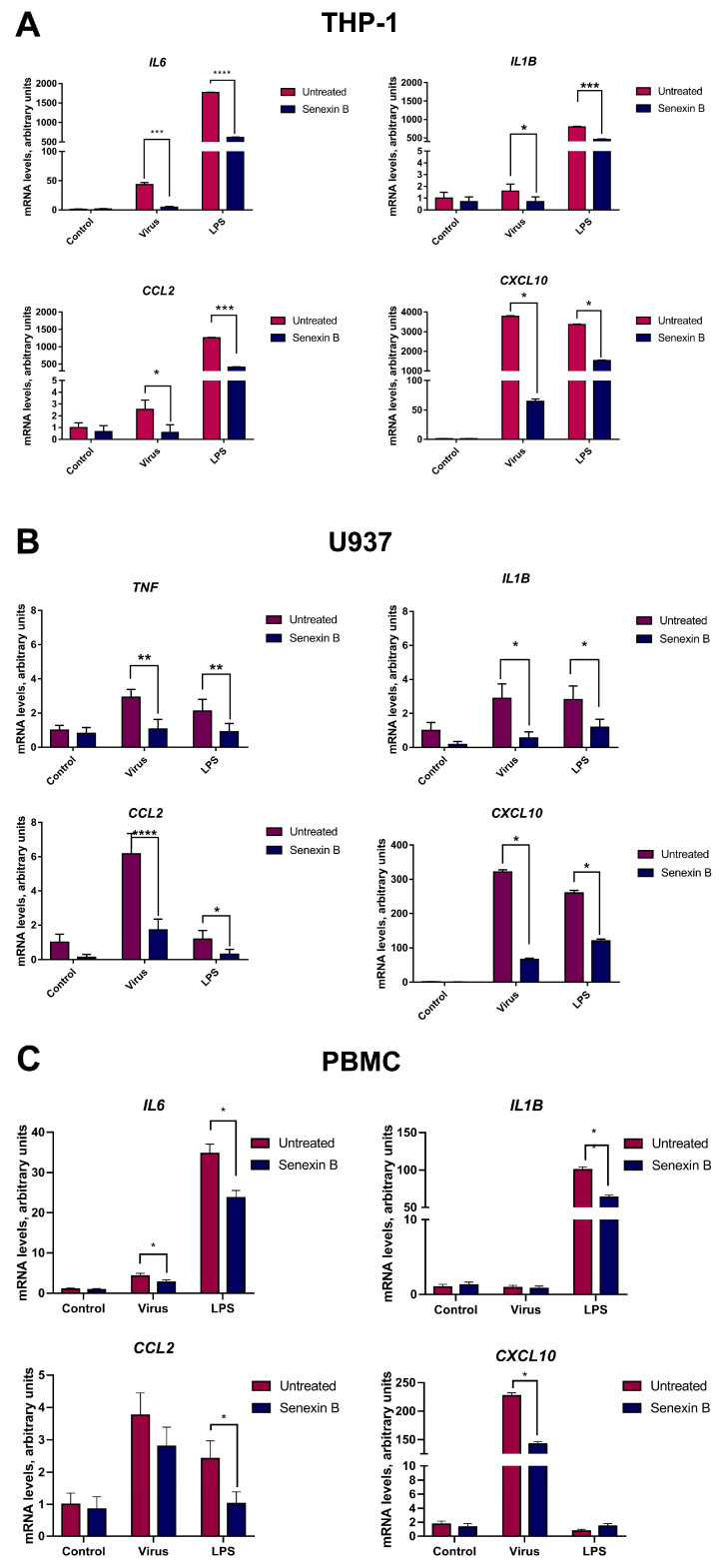
CDK8/19 inhibition reduces the induction of mRNA levels of key proinflammatory cytokines in monocytic cells. THP1, U937, and PBMCs were pretreated with 1 μM CDK8/19 inhibitor Senexin B for 1 h and then exposed to influenza virus or LPS for 24 h. (**A**–**C**) Relative mRNA levels in THP1 (**A**), U937 (**B**), and PBMCs (**C**). *HPRT* was used as a housekeeping control gene. Data are presented as mean ± SD of five independent biological experiments, each with two technical replicates. Differences between the groups were assessed using unpaired two-tailed *t*-test (* *p* < 0.05, ** *p* < 0.01, *** *p* < 0.001, **** *p* < 0.0001).

**Figure 2 viruses-15-01292-f002:**
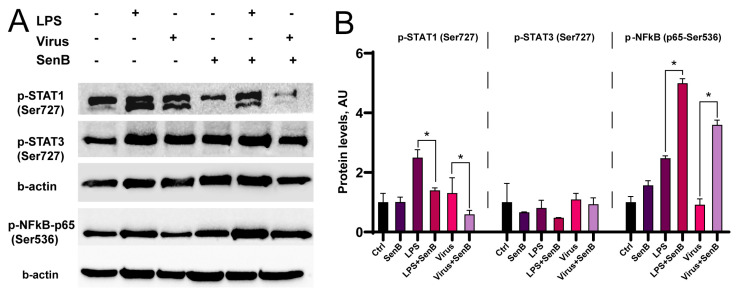
CDK8/19 inhibition reduces serine phosphorylation of STAT1 but not NFκB in proinflammatory conditions. (**A**) Representative images of Western blot data. Protein samples were collected from THP1 cells 24 h after exposure to LPS and 1 h pretreatment with Senexin B (SenB). Ctrl—Control; Virus—influenza A virus (A/Puerto Rico/8/34 (H1N1)). (**B**) Semiquantitative analysis of Western blot image. The protein area and signal intensity were analyzed using ImageJ software and normalized to β-actin level (Experiment 1: p-STAT, p-STAT3, β-actin; Experiment 2: p-NFkB-p65, β-actin). * *p* < 0.05.

**Figure 3 viruses-15-01292-f003:**
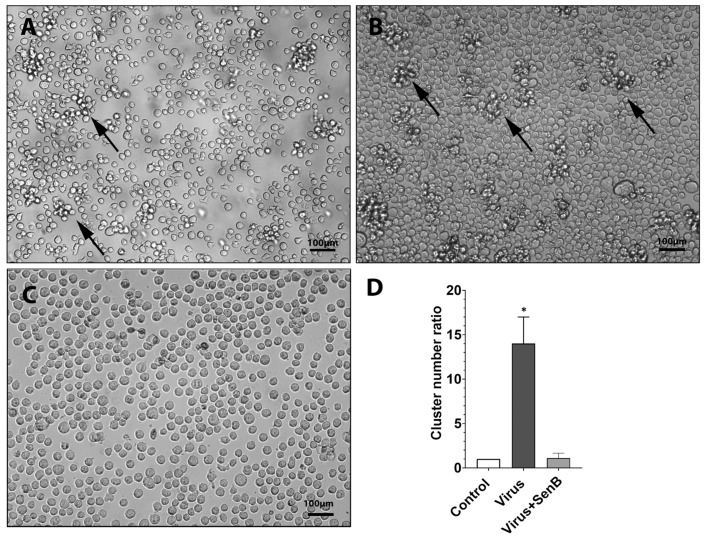
CDK8/19 inhibitor Senexin B decreases virus-induced cell clustering. (**A**–**C**) Images of untreated THP1 cells (**A**), cells treated for 24 h with virus influenza A virus A/Puerto Rico/8/34 (H1N1) alone (**B**) or virus with Senexin B (**C**). Cell aggregates are shown with arrows. (**D**) Clusters of THP1 cells exposed to the virus in the absence or presence of Senexin B (SenB). Average number of clusters in untreated cells was taken as 1. Data are mean ± SD of two independent biological experiments with eight replicates per group. * *p* < 0.05.

**Figure 4 viruses-15-01292-f004:**
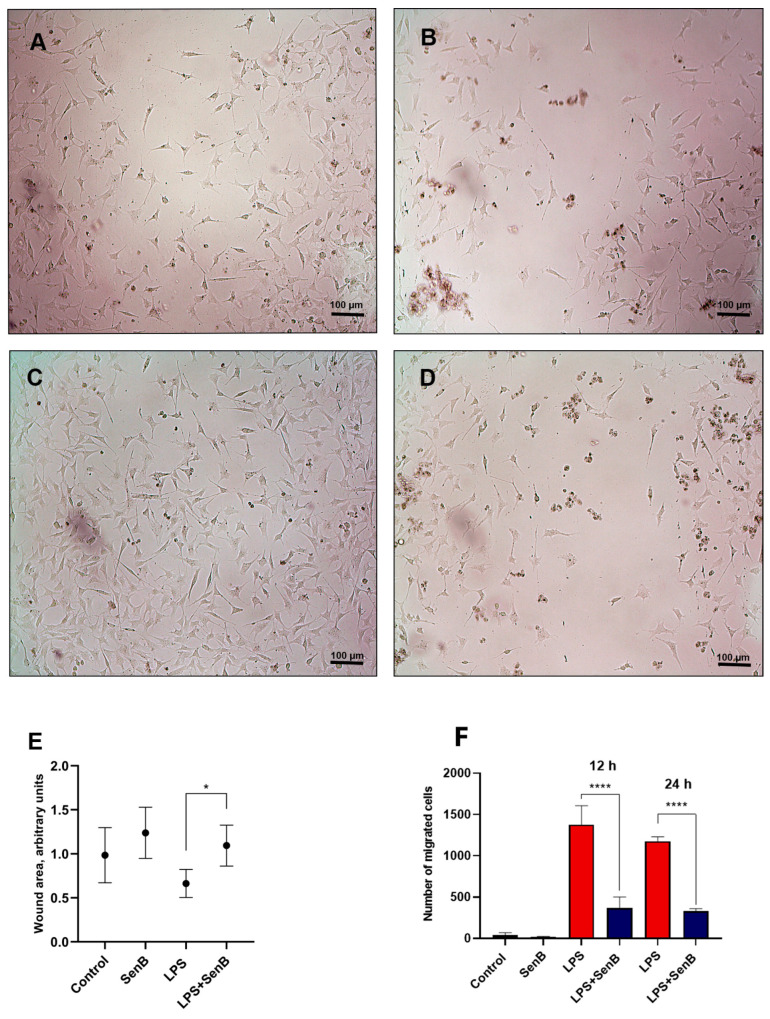
CDK8/19 inhibitor Senexin B decreases migration activity of human fibroblasts activated by LPS and downregulates migration-stimulating factors produced by LPS-treated THP1 monocytes. (**A**–**D**) Images of untreated primary human pulmonary fibroblasts (HPF) (**A**), cells treated for 24 h using Senexin B (**B**), LPS alone (**C**), or LPS with Senexin B (**D**). (**E**) Wound areas were measured using ImageJ program. Data obtained in 3 independent experiments, 3 replicates in each with 3–5 photos and 3–5 measurements per photo. The average of control is normalized to 1 for each experiment, then the data are combined (arbitrary units). Data are mean of normalized units ± SD. * *p* < 0.05. (**F**) Number of migrating THP1 cells exposed to conditioned medium from LPS- and Senexin B-treated THP1 monocytes. Cell culture medium enriched with migration stimulating secreted factors was collected from THP1 cells treated with LPS and Senexin B for 24 h. Data are mean ± SD of two independent biological experiments with eight replicates per group. **** *p* < 0.0001.

**Figure 5 viruses-15-01292-f005:**
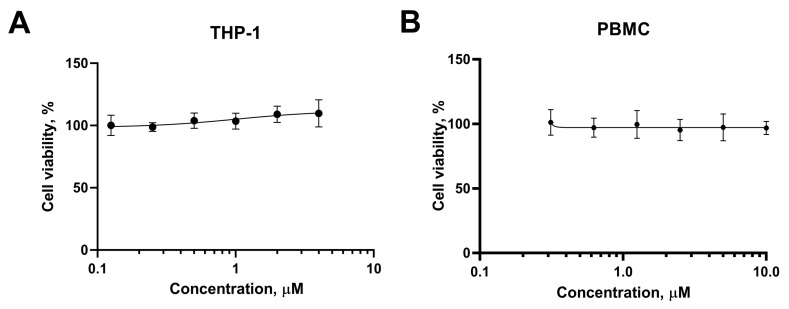
CDK8/19 inhibition does not interfere with cell viability or anti-inflammatory responses in monocytic cells. (**A**,**B**) Briefly, THP1 cells (**A**) or PBMCs (**B**) were incubated with the various concentrations of CDK8/19 inhibitor Senexin B for 72 h. Then, the cell viability was measured using a resazurin assay. Three biological experiments with three technical replicates per group were performed.

**Table 1 viruses-15-01292-t001:** Primer sequences used for real-time quantitative PCR.

Gene	Forward Primer Sequence (5′ > 3′)	Reverse Primer Sequence (5′ > 3′)
*IL6*	ATGAGGAGACTTGCCTGGTG	GCATTTGTGGTTGGGTCAGG
*IL1B*	TGGAAGGAGCACTTCATCTGTT	TCGCCAGTGAAATGATGGCT
*IL10*	CGCCTTGATGTCTGGGTCTT	CGAGATGCCTTCAGCAGAGT
*TNFA*	TCAGCCTCTTCTCCTTCCTG	GCCAGAGGGCTGATTAGAGA
*CXCL10*	AAGTGGCATTCAAGGAGTACCT	GGACAAAATTGGCTTGCAGGA
*CCL2*	TCTCAAACTGAAGCTCGCAC	CATTGATTGCATCTGGCTGAG
*HPRT1*	TATATCCAACACTTCGTGGGGTC	ACAGGACTGAACGTCTTGCTC

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
