# Peer review of "Inhibition of Cyclin-Dependent Kinases 8/19 Restricts Bacterial and Virus-Induced Inflammatory Responses in Monocytes"

_viruses, 2023, doi:10.3390/v15061292_

Round 1
Reviewer 1 Report
I do not think that the presented b actin control comes from the same experiment - please provide all gels for all experiment repetitions
This is only in vitro cell culture study
There are no limitations discussed
Some phosphorylation inhibitors should be used to strengthen the findings
Only a few proteins from very complex biological pathways were analyzed - how did the Authors choose these particular proteins?
Figures quality is very low making them unable to read
Please provide the full/uncut gels as a supplementary file - there are missing now
Partially English looks like translated using automatic translators - it is a mix of AmEn and British English.
Author Response
The authors are thankful to the reviewer for valueable comments, which definitely make paper more relevant. Please see the attachment with our responses for details.

Reviewer 2 Report
The search for compounds that reduce systemic inflammation caused by viruses and bacteria is relevant and is being performed very actively. The authors showed that a selective inhibitor of cyclin-dependent kinases 8/19, Senexin B, reduces the expression of proinflammatory cytokines and chemokines in two monocyte lines and in human PBMC. The effect of the inhibitor on other functional manifestations of inflammation was shown only on THP 1 cells. However, the mechanisms of action of Senexin B may differ in tumor and normal mononuclear cells. The lack of relevant data using PBMC reduces the value of the work.
Remarks:
1. Page 4, lines 171-178, Fig.1:
Data on the expression level of TNFÉ‘ mRNA in PBMCs and THP1 cells are not given, and nothing is said about it in the text.
It is necessary to give the levels of mRNA expression of the genes of the studied cytokines in control cells in order to compare cell lines for cytokine production.
2. Page 7, lines 206-207, Fig. 2:
It is written that "LPS increased the phosphorylation levels of STAT1 (Ser727), STAT3 (Ser727) and NFkB (Phospho-NFkB p65 (Ser536) (Fig. 2A, B)". However, the data on STAT3 does not confirm this statement.
Denote the Figures A and B.
3. Pp. 8-9, lines 229-233, Fig. 3:
Figure 3E is missing, Figures 3E and 3D are mixed up.
Author Response

(The authors gave the same response as above.)

Reviewer 3 Report
This manuscript by Elena Kokinos and co-workers addresses to determine the functions of cyclin-dependent kinases 8 and 19 in inflammatory responses induced by bacterial lipopolysaccharide and by the Influenza virus strain A/PR8 in THP1 and U937 monocytic cell lines and in peripheral blood-derived mononuclear cells and to evaluate the potential of senexin B, a selective quinazoline inhibitor of Cyclin-dependent kinases 8 and 19, to control pro-inflammatory activity. Results obtained demonstrated that senexin B is able to prevent the induction of inflammatory cytokine gene expression in all cell types examined and also reduced the functional manifestations of inflammation.
General comments:
The study is not particularly original as it is already known that therapies targeting CDK8/19 enzyme activity show promise for fighting virus infection and cancer and that inhibition of CDK8/19 suppresses the replication of other viruses or the inflammation triggered by the infection.
However, the paper is well structured. The title clearly indicates the focus of the article and the Abstract section efficiently summarizes the contents of the paper. Materials and methods” are suitable, the statistical analyses are well defined and appropriate to the design and Figures and Table are all necessary for understanding of the text.
The "Discussion" should be improved to explain the novelty of the results obtained and impact on infection control.
Minor comments:
2. Materials and Methods
2.1. Cell culture studies
Line 81: please substitute 100 μg/ml penicillin-streptomycin with 100 units/mL penicillin and 100 µg/mL streptomycin.
Line 82: Please change “The influenza virus strain A/Puerto Rico/8/34 (H1N1)” with “The Influenza A virus (A/Puerto Rico/8/1934(H1N1))” or “Influenza A virus (A/PR 8/34 (H1N1))”
2.3 Transwell migration assay
Line 102: substitute 1 uM Senexin B with 1μM Senexin B
2.4 Scratch Assay
Line 114: substitute 1 uM Senexin B with 1μM Senexin B
2.5 RNA extraction, reverse transcription and quantitative real-time PCR
Line 125: substitute 1 uM with 1μM
3. Results
3.1. CDK8/19 inhibition reduces the induction of mRNA levels of key pro-inflammatory cytokines by influenza virus and LPS in monocytic cells
Line 168: please change “influenza virus H1N1 (strain A/Puerto Rico/8/34)” with Influenza A virus (A/PR/8/34(H1N1))
Lines 180-182: The sentence “Senexin B had little-to-no impact on cytokine gene expression in quiescent cells, as mRNA levels of IL6, CXCL10 and TNFA did not change, while IL1B and CCL2 were slightly reduced in THP1, U937 or PBMCs (Fig. 1).” is incorrect as IL1B and CCL2 were statistically significantly reduced in U937 cells (****p < 0.0001). Please rewrite the sentence.
Concerning Figure1: Why hasn't TNF alpha been measured in THP1 and peripheral blood-derived mononuclear cells? Please add this information.
3.3. Reduced THP1 cell clustering and migratory activity in the presence of CDK8/19 inhibitor
Line 224: Please delete “the”. Change “Similarly, to the other viruses, viral strain A/Puerto Rico/8/34 (H1N1)…” with Similarly to other different viruses, Influenza A virus (A/PR/8/34(H1N1))…..
Line 229: Please change (Fig. 3A-D) with (Fig. 3A-C).
Lines 232 and 233: Please change (Fig. 3E) with (Fig. 3D).
4. Discussion
Line 332: Influenza A virus (A/PR/8/34(H1N1)) is preferred to influenza virus H1N1 (strain A/Puerto Rico/8/34) (line 241
Minor editing of English language required
Author Response

(The authors gave the same response as above.)

Round 2
Reviewer 1 Report
Thank you for your replies and improvements incorporated in the manuscript body.
The only remaining issue is western blot raw data. I do not see correspondence between representative blots and data shown on their graphs. This experiment has been done at least in 3 independent experiments (lines 151-152) so I would like to see each gel along with its calculation for densitometric analysis.
Author Response
Thank you for your comment. We included all the images of western blot in supplementary file and densitometric analysis in separate file.
Reviewer 2 Report
Page 6, lines 215-218: This insert should be moved to section 3.1. "CDK8/19 inhibition reduces the induction of mRNA levels of key pro-inflammatory cytokines by influenza virus and LPS in monocytic cells"
Author Response
Thank you for your valuable comment. We agree with your proposal and move these lines to the section 3.1.